# An intact C-terminal end of albumin is required for its long half-life in humans

Jeannette Nilsen[1,2], Esben Trabjerg[3], Algirdas Grevys[1,2], Claudia Azevedo[4], Stephen O. Brennan[5], Maria Stensland[1,6], John Wilson [7], Kine Marita Knudsen Sand[1,8], Malin Bern[1,2], Bjørn Dalhus[9], Derry C. Roopenian[7], Inger Sandlie[1,8], Kasper Dyrberg Rand [10] & Jan Terje Andersen[1,2 ✉]

Albumin has an average plasma half-life of three weeks and is thus an attractive carrier to improve the pharmacokinetics of fused therapeutics. The half-life is regulated by FcRn, a cellular receptor that protects against intracellular degradation. To tailor-design the therapeutic use of albumin, it is crucial to understand how structural alterations in albumin affect FcRn binding and transport properties. In the blood, the last C-terminal residue (L585) of albumin may be enzymatically cleaved. Here we demonstrate that removal of the L585 residue causes structural stabilization in regions of the principal FcRn binding domain and reduces receptor binding. In line with this, a short half-life of only 3.5 days was measured for cleaved albumin lacking L585 in a patient with acute pancreatitis. Thus, we reveal the structural requirement of an intact C-terminal end of albumin for a long plasma half-life, which has implications for design of albumin-based therapeutics.

[1] Centre for Immune Regulation (CIR) and Department of Immunology, Oslo University Hospital Rikshospitalet and University of Oslo, Norway, Oslo, Norway. [2] Institute of Clinical Medicine and Department of Pharmacology, University of Oslo, Oslo, Norway. [3] Institute of Molecular Systems Biology, Department of Biology, ETH Zurich, Zurich, Switzerland. [4] i3S, Instituto de Investigação e Inovação em Saúde, and Instituto de Engenharia Biomédica (INEB) and Instituto de Ciências Biomédicas Abel Salazar, Universidade do Porto and University of Porto, Porto, Portugal. [5] Molecular Pathology Laboratory, Canterbury Health Laboratories, Christchurch Hospital, Christchurch, New Zealand. [6] Proteomics Core Facility, Oslo University Hospital Rikshospitalet, Oslo, Norway. [7] The Jackson Laboratory, Bar Harbor, ME, USA. [8] Department of Biosciences, University of Oslo, Oslo, Norway. [9] Department of Medical Biochemistry and Department for Microbiology, Oslo University Hospital Rikshospitalet and University of Oslo, Oslo, Norway. [10] Protein Analysis Group, Department of Pharmacy, University of Copenhagen, Copenhagen, Denmark. ✉email: j.t.andersen@medisin.uio.no

Albumin acts as a versatile carrier of a wide range of substances, such as fatty acids, steroids, thyroid hormones, and metals, and plays an important role in stabilizing the extracellular fluid volume[1]. It is expressed by hepatocytes in the liver as a single polypeptide of 585 amino acids, and consists of 67% α-helices, which fold into three homologous domains denoted DI, DII, and DIII[2,3]. In the blood, it is the most abundant protein (40 g/L), with a long plasma half-life of ~3 weeks in humans[1].

Albumin is of interest for the pharmaceutical industry because of its ability to bind a wide range of small chemical drugs that consequently affect their pharmacokinetics[4]. In addition, it is increasingly utilized to improve the pharmacokinetics of short-lived bioactive therapeutic peptides and proteins by genetic fusion of such molecules to the N- or C-terminal end[5]. Two human serum albumin (HSA) fused drugs have been approved for clinical use; Tanzeum® and Idelvion®, which contain glucagon-like peptide 1 and recombinant coagulation factor IX, respectively. Both drugs are genetically fused to the N-terminal of HSA, which prolongs the half-life from 2 min to 5 days for the peptide and from 22 h to 102 h for the coagulation factor, giving rise to better dosing options for patients with type II diabetes and hemophilia B[6,7].

The biological mechanism responsible for its persistence in blood was revealed when Chaudhury et al.[8] discovered that albumin binds a cellular receptor, namely the neonatal Fc receptor (FcRn). At the time, FcRn was already known to bind antibodies of the IgG class and play a key role in controlling their long serum half-life[9–11]. It was then discovered that FcRn-deficient mice not only had dramatically reduced serum levels of IgG but also less than half of normal albumin levels. Furthermore, the half-lives of both proteins were considerably shortened in such mice compared with wild-type (WT) mice[8]. The cellular mechanism responsible has been studied by live-cell imaging tracking co-transport of IgG and FcRn in human endothelial cells, which shows that FcRn is predominantly found within acidic endosomes, where it binds IgG that has entered the cell by fluid-phase pinocytosis. Binding is only initiated at slightly acidic pH (5.0–6.5) and when the FcRn–IgG complex is recycled back to the cell surface membrane, exocytosis, and exposure to the physiological pH (7.4) of the blood triggers dissociation and release of IgG back into the circulation[12–17]. Thus, this binding-and-release mechanism is based on strict pH-dependent binding between FcRn and IgG. Similarly, FcRn binds albumin at slightly acidic pH and not at neutral pH, and recent reports support that albumin is recycled via the same route, independently of IgG[18–21].

FcRn belongs to a family of major histocompatibility class I-related molecules, and consists of a unique heavy chain (HC), with three extracellular domains (α1, α2, and α3) in non-covalent association with β2-microglobulin (β2m)[22–24]. Albumin binds to a region on FcRn that is distal from the IgG-binding site, and site-directed mutagenesis studies as well as X-ray crystallographic studies have provided in-depth insight into how FcRn binds albumin in a pH-dependent fashion[8,25–27]. At slightly acidic pH, H166 within the α2-domain of the HC of FcRn stabilizes a loop in the α1-domain through intramolecular interactions with E54 and Y60, which are not formed at neutral pH[25,28–30]. This makes conserved tryptophan residues (W53 and W59) in the loop available for binding to albumin[29]. DIII of HSA contains the main binding site for human FcRn (hFcRn), where three conserved histidine residues (H464, H510, and H535) sustain hydrophobic pockets within DIII in a conformation that promotes insertion of W53 and W59 of hFcRn at slightly acidic pH[26,30–32]. More recently, we and others demonstrated that DI of HSA modulates the interaction via two surface-exposed loops that form direct contacts with the receptor[26,31,33].

The discovery that FcRn is a key controller of albumin homeostasis opens new possibilities for rational design of albumin-based therapeutics, such as to tailor the half-life of drugs by engineering the interaction. One of the first examples of such engineering is an HSA variant with a single-point mutation in DIII (K573P), which improves binding to hFcRn by more than tenfold and extends the plasma half-life from 5.4 to 8.8 days in cynomolgus monkeys[34]. The variant is one of a panel of engineered HSA variants with altered pharmacokinetics that together constitute the so-called Veltis® technology.

To further optimize the therapeutic use of albumin, it is crucial to fully understand how structural alterations and fusion to albumin affects its ability to bind and to be transported by FcRn. In this study, we demonstrate that truncations and amino acid substitutions in the last C-terminal α-helix of albumin strongly reduce binding to the receptor, where even removal of the last residue (L585) has a negative effect on FcRn binding. HDX-MS analysis shows that deletion of L585 causes structural stabilization in regions of the C-terminal DIII that otherwise require flexibility to allow optimal binding. Furthermore, lack of L585 resulted in reduced rescue from intracellular degradation and a shorter serum half-life in hFcRn transgenic mice. In agreement, we found that the truncated variant was quickly eliminated from the blood of a patient diagnosed with acute gallstone-induced pancreatitis and measured a half-life of 3.5 days only. Moreover, exposure to carboxypeptidase A, an enzyme found at elevated levels in plasma of individuals with pancreatitis, resulted in cleavage of L585 from the C-terminal end of HSA, also in the approved drug Tanzeum®, and consequently reduced hFcRn binding. Thus, we provide evidence for the requirement of a structurally intact C-terminal end of HSA to maintain optimal binding to hFcRn and a long plasma half-life, which should guide design of the next generation of albumin-based therapeutics.

## Results

**C-terminal modified HSA variants show reduced FcRn binding.** The three homologous domains (DI, DII, and DIII) of albumin are connected by flexible loops and form a heart-shaped structure, where DIII harbors the principal binding site for FcRn[2,3,26,30]. Interestingly, we have observed that introducing mutations in the last C-terminal helix of DIII affects binding to hFcRn, which is not easily explained by available co-crystal structural models[26,31,34]. We became aware of four naturally occurring HSA variants with alterations in this C-terminal region, which are named Bazzano, Catania, Rugby Park, and Venezia[35–38]. Bazzano and Catania are products of a single-nucleotide deletion and a subsequent frameshift, resulting in truncated variants lacking three amino acids and with replacement of the last 19 or 6 C-terminal amino acid residues, respectively[36,38] (Fig. 1a). Rugby Park and Venezia are truncated by seven amino acids, and have their C-terminal ends altered from position 572, due to a splice error in intron 13 (Rugby park) or a 30-bp deletion and a 5-bp insertion resulting in skipping of exon 14 (Venezia)[35–37] (Fig. 1a).

We produced recombinant versions of the variants and observed only minor differences in expression levels in comparison with WT HSA (Fig. 1b, c). To measure receptor binding, surface plasmon resonance (SPR) measurements were conducted by immobilizing the HSA variants followed by injecting titrated amounts of monomeric hFcRn at pH 5.5. Dissociation constants ($K_D$) were calculated by fitting the sensorgrams to a simple 1:1 Langmuir binding model or a steady-state affinity-binding model. All four variants bound the receptor less well than the WT, where Catania and Venezia showed a 6.6- and 7.6-fold

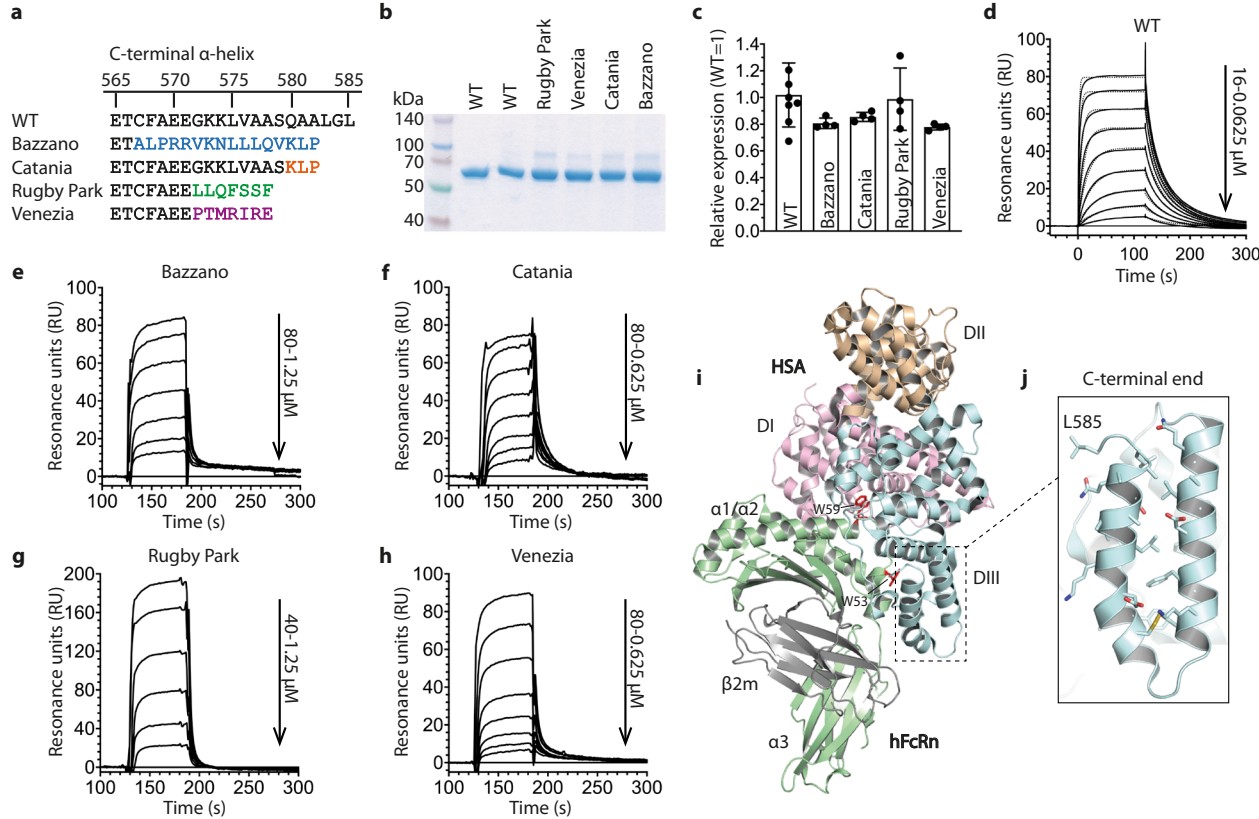

**Fig. 1 Genetic HSA variants with modified C-terminal ends show reduced FcRn binding. a** An overview of the C-terminal sequences of genetic HSA variants. **b** A 12% SDS-PAGE gel stained with Coomassie Blue showing 2 μg of purified recombinant WT HSA and genetic variants. The gel was cropped and the full-length gel is shown in Supplementary Fig. 1. **c** Relative expression levels of WT HSA and genetic variants by transiently transfected HEK293E cells. Protein secreted into the growth medium was quantified by ELISA, and the values represent the mean ± s.d. of four replicates. **d** SPR sensorgrams showing binding of titrated amounts (0, 0.0625, 0.125, 0.25, 0.5, 1, 2, 4, 8, and 16 μM) of monomeric hFcRn injected over immobilized (~200 RU) WT HSA at pH 5.5 (—) and the fit of the data to the 1:1 binding model (·····). Injections were performed at 25 °C, and the flow rate was 40 μl/min. **e–h** SPR sensorgrams showing binding of titrated amounts (0, 0.625, 1.25, 2.5, 5, 10, 20, 40, and 80 μM) of monomeric hFcRn injected over immobilized (~500 RU) (**e**) Bazzano, (**f**) Catania, (**g**) Rugby Park, and (**h**) Venezia at pH 5.5. Injections were performed at 25 °C, and the flow rate was 10 μl/min. (**i**) An illustration of the co-crystal structure of WT HSA in complex with hFcRn. DI, DII, and DIII of HSA are colored in pink, wheat, and light blue, respectively. The three domains, α1, α2, and α3, of the HC of hFcRn are shown in light green and the β2m subunit in gray. **j** A close-up of the two last α-helices of the C-terminal end of HSA. The figures were made using PyMOL with the crystallographic data of WT HSA-hFcRn (PDB ID 4N0F)[26].

**Table 1 SPR-derived kinetics for binding of HSA variants to hFcRn.**

| HSA variant[a] | $k_a$ (10⁴ M⁻¹ s⁻¹) | $k_d$ (10⁻² s⁻¹) | $K_D$[b] (μM) | $K_D$[c] (μM) |
|---|---|---|---|---|
| WT | 4.8 | 3.3 | 0.7 | 1.1 |
| L585X | 1.5 | 5.6 | 3.7 | 5.3 |
| Bazzano | NA[d] | NA | NA | 10.0 |
| Catania | NA | NA | NA | 7.3 |
| Rugby Park | NA | NA | NA | 11.3 |
| Venezia | NA | NA | NA | 8.4 |

[a]The HSA variants were immobilized on CM5 chips and serial dilutions of hFcRn were injected at pH 5.5.
[b]The kinetic rate constants were obtained using a simple first-order (1:1) Langmuir bimolecular interaction model.
[c]The steady state affinity constant was obtained using an equilibrium (Req) binding model.
[d]NA, not acquired due to fast kinetics.

decrease in affinity, respectively, while Bazzano and Rugby Park bound with 9–10-fold weaker affinity (Fig. 1d–h, Table 1).

As an attempt to explain the effects of the naturally occurring C-terminal alterations on receptor binding, we scrutinized a co-crystal structural model of WT HSA in complex with hFcRn[26] (Fig. 1i). None of the residues of the last C-terminal α-helix of HSA contact the receptor, but several of them form intramolecular interactions with the second last α-helix (Fig. 1j). Both α-helices are part of the hydrophobic pocket that accommodates W53 of FcRn, where the second last α-helix contains residues (e.g., F551) that form hydrophobic interactions with the tryptophan[26]. Thus, truncating and substituting amino acid residues of the last C-terminal α-helix, as in the mutated variants, probably disrupt some of the helix–helix interactions and consequently the binding pocket, resulting in the negative effect on hFcRn binding.

**L585 is required for optimal FcRn binding.** Survey of albumin sequences from 20 species indicates that all lack the C-terminal L585 found in albumin from orangutan, chimpanzee, and human (Fig. 2a). Moreover, clinical data demonstrate that 4–15% of circulating albumin in humans also lack this residue[39,40]. To address the effect of loss of L585, we expressed a truncated recombinant version of HSA (L585X) (Fig. 2b). Circular dichroism (CD) analysis at 25 °C and pH 5.5 showed very similar ellipticity spectra as that of the WT (Fig. 2c). Binding to hFcRn at pH 5.5 and pH 7.4 was first compared by ELISA, where the receptor was captured onto a mutated hIgG1 variant that binds strongly to FcRn at both pH conditions[41]. The HSA variants were

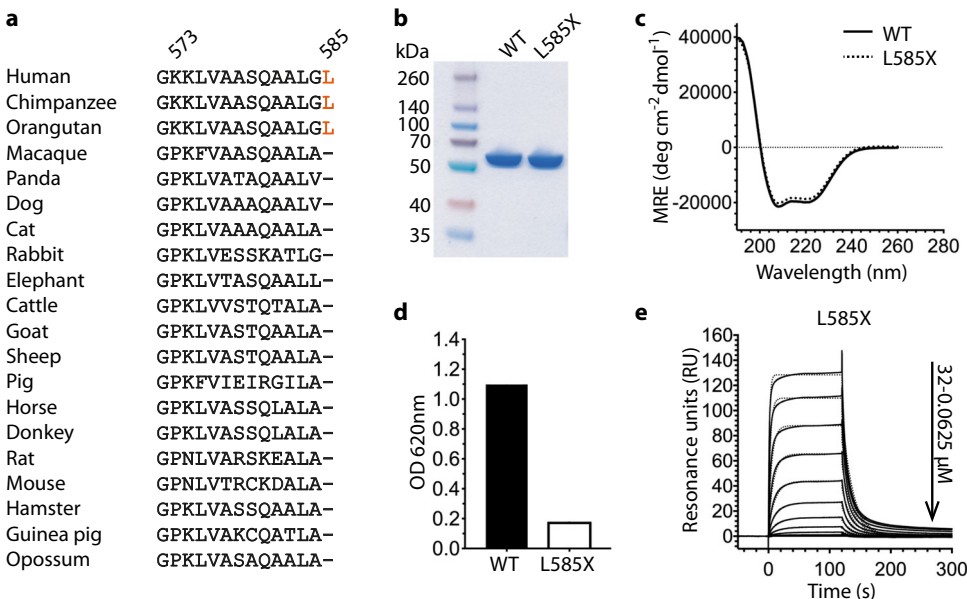

**Fig. 2 Removal of L585 from HSA causes reduced hFcRn binding. a** An alignment of the last 14 or 15 amino acid residues of the C-terminal α-helix of multiple albumin species. **b** A 12% SDS-PAGE gel stained with Coomassie Blue showing 2 μg of purified recombinant WT HSA and L585X. **c** CD spectra of WT HSA and L585X obtained at 20 °C and pH 5.5. MRE mean residual ellipticity. **d** ELISA showing binding of equal amounts (225 nM) of WT HSA and L585X to hFcRn at pH 5.5. **e** SPR sensorgrams showing binding of titrated amounts (0, 0.0625, 0.125, 0.25, 0.5, 1, 2, 4, 8, 16, and 32 μM) of monomeric hFcRn injected over immobilized (~300 RU) L585X at pH 5.5 (—) and the fit of the data to the 1:1 binding model (·····). Injections were performed at 25 °C and the flow rate was 40 μl/min.

then added and subsequently detected using polyclonal anti-HSA from goat. Importantly, the detection antibody cannot bind via its Fc to FcRn, as the IgG-binding site is already occupied when added. Moreover, as the binding sites for IgG and albumin are located at opposite sides of the receptor, the albumin-binding site should be readily available. The result showed weaker binding of the truncated variant compared with the WT at pH 5.5, while binding of neither of the two was detected at pH 7.4 (Fig. 2d; Supplementary Fig. 2a). We then extended the analysis to SPR, where L585X was immobilized, and titrated amounts of monomeric hFcRn were injected at pH 5.5. Fitting the binding curve to a 1:1 Langmuir binding model revealed a 5.3-fold reduction in binding affinity, as a result of slower on-rate and faster off-rate, for L585X compared with the full-length (Fig. 2e, Table 1). As expected, no binding response was obtained when hFcRn was injected at pH 7.4 (Supplementary Fig. 2b, c). Thus, removal of the last C-terminal amino acid residue of HSA reduces binding to hFcRn.

**Deletion of L585 causes structural stabilization in DIII.** As the HSA-hFcRn co-crystal structure does not directly explain the effect of removing L585, we studied the structural dynamics of HSA with and without the C-terminal leucine using hydrogen/deuterium exchange mass spectrometry (HDX-MS) at pH 5.5. The rate at which backbone amides exchange their protons with deuterium from solvent provides information about the conformational dynamics of proteins, as the rate of exchange of backbone amides is primarily determined by the stability of local hydrogen bonding[42–44]. The exchange rate can vary up to seven orders of magnitude, when comparing a backbone amide in a heavily structured region, such as in stable α-helices and β-strands, to a backbone amide in less structured or disordered regions (non-hydrogen bonded)[43,45]. Briefly, HDX was initiated by diluting WT HSA or L585X into deuterated buffer. The exchange reaction was stopped at various time points from 15 s up to 2960 min (i.e., sampled across a time range of

approximately four orders of magnitude) by lowering the temperature and pH. The quenched samples were digested by pepsin, before the mass of the resulting peptides were determined by liquid chromatography–mass spectrometry (LC–MS). Deuterium incorporation for 77 peptides was followed for both the WT and the truncated variant, covering 91% of the sequence (Supplementary Figs. 3 and 4). Most peptide segments of HSA did not display a difference in HDX between WT and L585X (Fig. 3e; Supplementary Fig. 3). Also, no difference was observed for the peptides corresponding to the last part of the C-terminal α-helix, when correcting for the difference in the "experimental maximally" labeled samples (Fig. 3a). However, the HDX differed in three other regions in DIII; residues 409–420 (Fig. 3b), 496–506 (Fig. 3c), and 530–550 (Fig. 3d), which are spatially in close proximity to the C-terminal end in crystal structures of HSA alone as well as when in complex with hFcRn (Fig. 3f, g). The significance of these reductions in HDX was further supported in the data from several overlapping peptides comprising these regions of DIII (Supplementary Fig. 3). Interestingly, for all three identified regions, the rate of HDX was slower for L585X, which suggests that these regions are less dynamic in the truncated variant. Thus, deletion of L585 causes local structural stabilization in distinct regions of DIII.

**HSA lacking L585 is more rapidly degraded.** To study the consequence of lack of L585 on FcRn-mediated recycling, we used the newly developed human endothelial cell-based recycling assay (HERA)[20]. Briefly, human microvascular endothelial cells (HMEC1) that overexpress hFcRn were grown in wells until 95–100% confluency. The cells were starved for 1 h, before equal amounts of WT HSA or L585X were added and incubated for 4 h to allow uptake. The cells were washed, and fresh pH 7.4 buffer was added. After another 4 h incubation, the amount of HSA released from the cells was quantified using ELISA, which revealed that only half the amount of the truncated variant was recycled compared with full-length HSA (Fig. 4a).

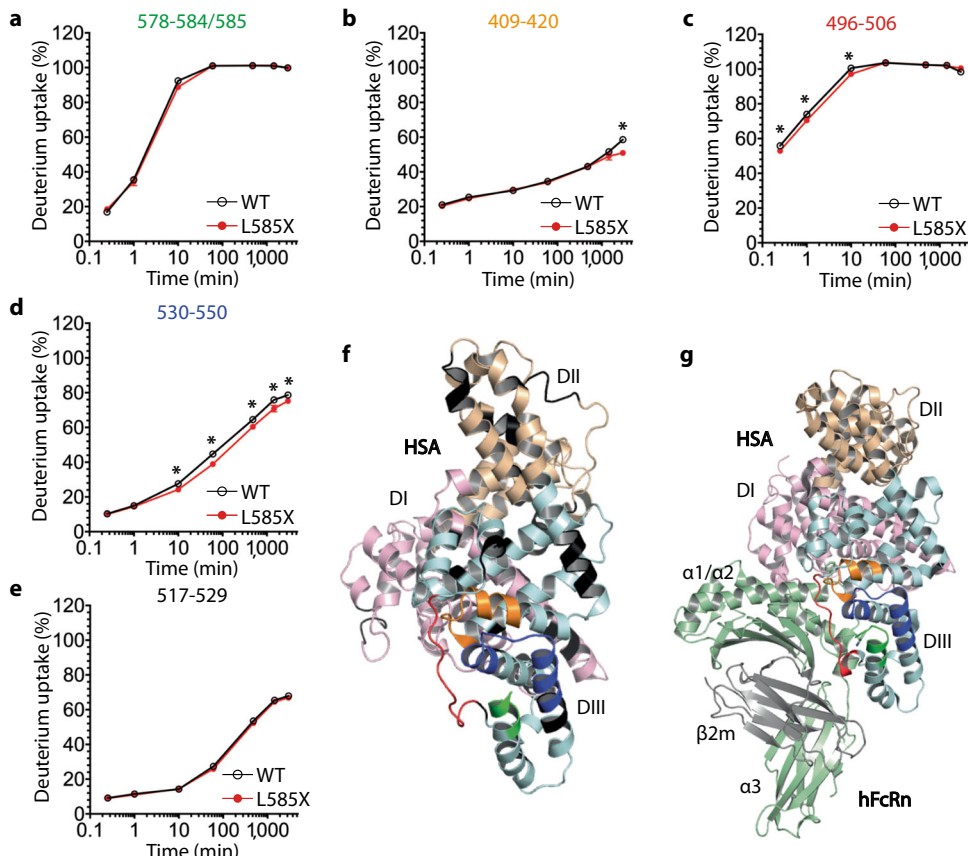

**Fig. 3 HDX-MS analysis shows structural stabilization in three regions of HSA lacking L585. a–e** Deuterium uptake plots for peptide segments of WT HSA (black) and L585X (red) of (**a**) the C-terminal peptides (i.e., with or without L585), **b–d** the peptide regions that showed a statistically significant difference in deuterium incorporation and (**e**) a peptide that showed no difference. The values were normalized to the deuterium uptake of the corresponding peptide in the "maximally labeled" control sample. The data points represent the mean ± s.d. of triplicates, except for 2960 min, where $n = 1$. *Marks a statistically significant change in HDX as defined in the "Methods" section. **f, g** The C-terminal peptide is highlighted in green, whereas the structural regions stabilized by removal of L585 are highlighted in orange (409–420), red (496–506), and dark blue (530–550) on the crystal structures of HSA (DI; pink, DII; wheat, DIII; light cyan) (**f**) alone and (**g**) in complex with hFcRn (HC; light green, $\beta_2m$; gray). The regions that were not covered in the HDX data are colored in black. The figures were made using PyMOL with the crystallographic data of HSA (PDB ID 1AO6) and the WT HSA-hFcRn co-complex (PDB ID 4N0F)[26].

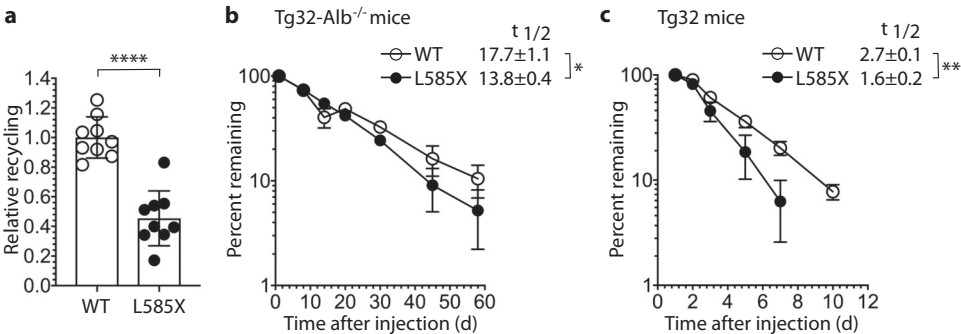

**Fig. 4 HSA lacking L585 shows decreased cellular recycling and serum half-life in mice. a** HERA showing relative cellular recycling of WT HSA and L585X. Equal amounts of each variant (1000 nM) were incubated with HMEC1 cells for 4 h, followed by extensive washing and another 4 h incubation step before sample collection. The amounts of recycled WT HSA and L585X were quantified by ELISA, and the obtained data are shown as mean ± s.d. of three independent experiments where $n = 3$. ****$p < 0.0001$, by unpaired Student's $t$ test. **b, c** Elimination curves of WT HSA (open circle) and L585X (filled circle) in hFcRn transgenic mice. Tg32-Alb$^{-/-}$ mice received 4 mg/kg HSA via intraperitoneal injection (IP) on day 0 ($n = 4$) (**b**). Tg32 mice received 2 mg/kg HSA via intravenous injection (IV) on day 0 ($n = 5$) (**c**). The serum levels of HSA are presented as percentage remaining in the circulation compared to day 1. The curves represent the mean ± s.d of four or five mice. Mean $\beta$-phase half-lives ($t_{1/2}$) ± s.e.m are shown in days. *$p = 0.0245$, **$p = 0.0023$, by unpaired Student's $t$ test.

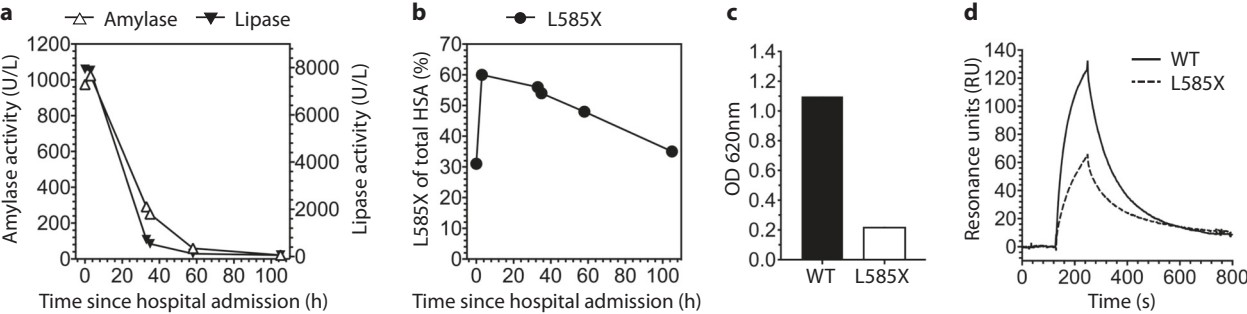

**Fig. 5 Plasma-derived L585X shows reduced binding to hFcRn and a short serum half-life in a patient with acute pancreatitis. a, b** Plasma activity of amylase (upward triangle) and lipase (downward triangle) (**a**) and the percentage of L585X (**b**) over time in blood of a patient with acute pancreatitis. The percentage of L585X was calculated from the mass spectra on the total HSA (Supplementary Fig. 5). **c** ELISA showing binding of equal amounts (225 nM) of WT HSA and plasma-derived L585X to hFcRn at pH 5.5. **d** SPR sensorgrams showing binding of equal amounts (1000 nM) of plasma-derived WT HSA and L585X injected over immobilized (~2000 RU) hFcRn at pH 5.5. Injections were performed at 25 °C, and the flow rate was 50 μl/min.

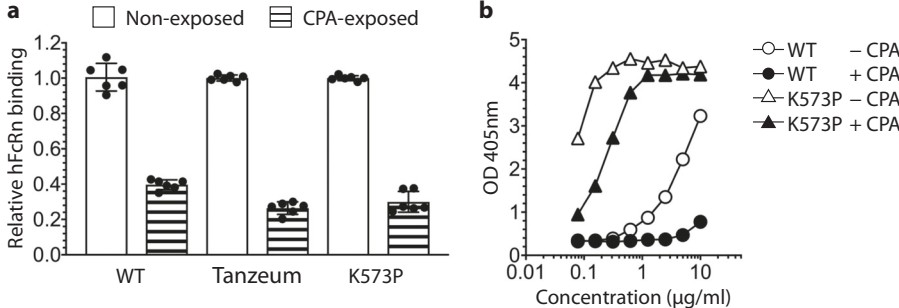

**Fig. 6 HSA exposed to CPA shows reduced hFcRn binding. a** Relative hFcRn binding at pH 5.5 of HSA variants exposed (striped bars) or nonexposed (white bars) to CPA. Calculated from ELISA results in (**b**) and in Supplementary Fig. 6, where nonexposed was set to 1. The data are shown as the mean ± s. d. of three independent experiments where $n = 2$. **b** ELISA showing binding to hFcRn at pH 5.5 of WT HSA (circle) and K573P (triangle) that had been exposed (filled black) or not exposed (filled white) to CPA.

Next, we measured the serum half-life of HSA with and without L585 in two humanized mouse models, the so-called Tg32 and Tg32-Alb$^{-/-}$ mouse strains[46,47]. While both strains express hFcRn instead of the mouse receptor, the latter also lacks expression of mouse albumin. In the albumin knockout mice (Tg32-Alb$^{-/-}$), a serum half-life of about 14 days was measured for L585X as compared with 18 days for the WT (Fig. 4b). In the presence of high levels of endogenous albumin (Tg32), the truncated variant was eliminated even more rapidly relative to full-length HSA, showing 1.7-fold shorter serum half-life (decreased from 2.7 to 1.6 days) (Fig. 4c). Notably, both variants showed monophasic log-linear decay curves, supporting the lack of immunogenicity and of an antibody response, even in the albumin deficient mice. The transgenic mice used have low levels of endogenous IgG due to weak binding affinity between hFcRn and mouse IgG[8,48–50]. However, engineered HSA variants injected into hFcRn transgenic mice have previously been shown to elicit antibody responses and cause increased clearance[31].

Thus, C-terminal truncation leads to less efficient cellular recycling and reduced serum half-life as a result of altered FcRn-binding kinetics. HSA lacking L585, which binds hFcRn with fivefold weaker affinity due to a slower on-rate and a faster off-rate, is thus less likely to bind the receptor in the endosomes, especially in the presence of large amounts of competing endogenous albumin, and instead follow the default pathway to the lysosomes.

**Truncated HSA lacking L585 has a short half-life in humans.** Patients with pancreatitis have elevated pancreatic protease

activity in the circulation and as a result, the level of HSA lacking L585 may increase dramatically[39,40]. We collected six blood samples from a patient with acute gallstone induced pancreatitis over a time period of 104 h, starting at the time the patient was admitted to the hospital. To monitor the condition, we measured the plasma activity of amylase and lipase (Fig. 5a), which are commonly used diagnostic markers of pancreatitis[51]. Plasma samples taken the first day showed markedly elevated activity of both amylase (974–1023 U/I) and lipase (7880–7926 U/I), which had decreased substantially by the 30 h blood sample (291 U/I and 714 U/I, respectively), supporting that the pancreas was recovering. Over the 104 h time period, the total serum albumin concentration decreased slightly from 40.9 g/L to 35.9 g/L (Supplementary Fig. 5a). The mass and heterogeneity of purified HSA were assessed using LC–ESI–MS, which showed that a considerable amount had a truncated C-terminal (Fig. 5b; Supplementary Fig. 5b). The level of L585X increased during the first 2.2 h, which is when the level peaked at 60% of the total HSA. Following the initial rise in concentration, the level declined over the next 102 h, from which a half-life of only 3.5 days was calculated (Fig. 5b). Moreover, we isolated truncated HSA from a patient with 100% L585X, while full-length HSA was isolated from a healthy individual in which L585X accounted for ~3% of the total albumin (Supplementary Fig. 5c). As measured for the recombinant version, plasma-derived L585X showed reduced hFcRn binding compared with full-length HSA (Fig. 5c, d).

**Carboxypeptidase A cleaves L585 of HSA.** Carboxypeptidase A (CPA), also referred to as CPA1, is a pancreatic protease that

cleaves aromatic and aliphatic amino acid residues from the C-terminal of proteins[52]. It is the enzyme predicted to cleave the C-terminal L585 of HSA[39,53]. This is supported by increased serum concentrations of CPA observed in parallel with elevated serum levels of L585X in patients with pancreatitis[54,55]. We incubated WT HSA with and without CPA purified from human pancreas at 37 °C for 5 h followed by MS analysis, which confirmed that exposure to the enzyme generated a product without L585 (Supplementary Table 1)[53]. The protocol was repeated with the commercially available N-terminal HSA-drug fusion, Tanzeum®. After exposure to CPA, more than 99% of the C-terminal peptides detected were lacking L585, whereas the leucine was present in the nonexposed sample (Supplementary Table 1). In addition, we included the engineered variant, HSA K573P, which has improved pH-dependent binding affinity for hFcRn and extended half-life in nonhuman primates[34]. Incubation with CPA resulted in complete removal of L585 from the C-terminal end of the engineered variant (Supplementary Table 1). The effect of losing L585 on hFcRn binding was then evaluated by ELISA, which demonstrated weaker binding of the enzyme-exposed HSA and fusion construct as compared with the nonexposed variants (Fig. 6a; Supplementary Fig. 6). The same was observed for HSA K573P, however, the cleaved version still bound more strongly compared with non-cleaved WT HSA (Fig. 6a, b). Thus, CPA cleaves the C-terminal L585 of HSA and causes reduced FcRn binding.

## Discussion

Albumin is the most abundant protein in blood and plays a key role as carrier of primarily small hydrophobic and insoluble molecules throughout the body. To take advantage of its long plasma half-life and improve in vivo efficacy of drugs, one can either target drugs to HSA via albumin-binding molecules, use site-specific conjugation to C34 of HSA or genetically fuse proteins of interest to the N- or C-terminal end of HSA[5]. In this context, knowledge about how FcRn binds and controls the plasma half-life and transport of albumin, as well as how structural alterations and fusion to albumin affect these properties, will improve design of novel albumin-based therapeutics with tailored pharmacokinetics.

We have previously reported that the C-terminal domain (DIII) of HSA contains the main binding site for hFcRn, and that DI modulates the interaction[30,33,56]. In addition, we have shown that deletion of the last α-helix of HSA reduces binding[30,34]. More than 70 genetic variants of HSA have been identified[57]. Here, we studied the Bazzano, Catania, Rugby Park, and Venezia variants, which have amino acid substitutions and truncations in the C-terminal end[35–38]. The variants were first identified during routine clinical electrophoresis run on serum from heterozygous carriers of one normal and one mutated albumin gene, in which the variants are present at lower levels than the normal protein[35,37,38]. Thus, they are secreted into the circulation, but it is not known whether the genetic modifications affect the rate of expression and secretion by liver cells. Furthermore, CD analysis has demonstrated that the modifications in the C-terminal end increase or decrease the overall α-helical content and thermal stability[58,59]. In this study, we found that all four variants have reduced binding affinity toward hFcRn. The weakest binding was measured for Rugby Park, which is present at only 8% of the total serum albumin, followed by Bazzano, Venezia, and Catania representing 18%, 30%, and 30%, respectively[35,37,38]. Thus, the FcRn-binding affinity of the variants correlate with their serum level in heterozygous carries.

Furthermore, we demonstrate that a fully intact C-terminal end of HSA is required for optimal binding to hFcRn, as removal of the very last L585 residue resulted in more than fivefold weaker affinity due to slower association and faster dissociation. HDX-MS data support that differences in structural dynamics contribute to reduced binding of the truncated variant, as deletion of the leucine led to structural stabilization of three regions in DIII, that have been reported to undergo large conformational changes upon receptor binding (up to 19 Å in Cα–Cα distances)[26]. The regions are located in or in close proximity to the interaction interface in the co-crystal structure, and it has previously been shown that targeted mutagenesis within these segments affect receptor binding[30,31]. The data support that the presence of L585 causes a functionally important local destabilization and structural flexibility within DIII that is in favor of hFcRn binding.

From an evolutionary perspective, it is interesting that the C-terminal leucine residue is not conserved, as it is only expressed by human, chimpanzee, and orangutan. These are also the only species that express albumin with a lysine instead of a proline in position 573, which also has great influence on FcRn binding[34,56]. Thus, in regard to binding affinity for hFcRn, the L585 residue compensates for the lack of P573 in HSA. Since transporting cargo is a main function, we speculate that different requirements to albumin in this regard may have been the driving force resulting in these species differences. Notably, as the leucine influences the structural dynamics of DIII of HSA, it may well be that its presence also affects the ligand binding properties of the domain.

Interestingly, HSA that lacks the last C-terminal L585 residue is present in the blood of healthy individuals, where it has been measured to constitute 4–15% of the circulating albumin pool[39,40]. In line with this, we found that ~3% of the total albumin in plasma from a healthy control lacked the leucine. Like the recombinant version, reduced binding to hFcRn was measured for L585X isolated from human plasma. As such, reduced receptor binding provides an explanation for its short in vivo half-life of 3.5 days. This was measured in a patient with acute gallstone-induced pancreatitis, which is a condition where the pancreatic duct that connects the pancreas to the small intestine is blocked, leading to premature activation of pancreatic enzymes that attack the pancreas[51]. At most, we found that 60% of the total albumin was lacking L585, which is in line with previous reports showing levels ranging from 17 to 93% in patients suffering from pancreatic disease[39,40]. The truncated form is most likely a result of exposure to CPA, which is synthesized by the pancreas and normally released as a zymogen into the gut, but found at elevated levels in plasma of individuals with pancreatitis[39,60,61]. Similarly, a tenfold increase in the plasma concentration of CPA, and a high percentage of circulating L585X, has been measured in a severely traumatized patient[62]. We speculate whether cleavage of L585 may be a way to promote the clearance of albumin that carries metabolic waste products that accumulate during acute disease. Interestingly, as the leucine is not conserved, this would not necessarily apply to all species. Future studies should address whether various ligands whose binding induce conformational changes in HSA, could make the protein more or less susceptible to enzymatic cleavage. Moreover, as the basal level of the activated form of CPA in plasma is very low[60], this raises questions as to where and when HSA is cleaved under normal conditions.

Our findings also have implications for HSA-based therapeutics with a free C-terminal end, as exemplified here with a N-terminal drug fusion approved for clinical use. Incubation of the drug with CPA resulted in reduced binding to hFcRn due to loss of L585. Thus, exposure to the enzyme in vivo will affect the pharmacokinetics. Instead, genetic fusion of a drug of interest to the C-terminal of HSA may be favorable to prevent enzymatic cleavage, as long as the drug does not represent a substrate for

CPA itself or compromise hFcRn binding. We previously demonstrated that genetic fusion of a small peptide or a single-chain-variable fragment to both the N- or C-terminal end of HSA slightly decreased binding to hFcRn[63]. Thus, it is crucial to address whether or not a particular fusion of interest interferes with binding to hFcRn, as it may vary depending on the nature of the fused protein. Importantly, to secure long serum half-life, engineered HSA variants with improved binding to hFcRn, such as the K573P variant[34], may be used to compensate for diminished receptor binding caused by C-terminal truncation or design.

## Methods

**Construction and production of HSA variants.** A pcDNA3.1 vector containing cDNA that encodes for full-length HSA has been described previously[20], and cDNA fragments encoding modified DIII sequence (K573P, L585X, Bazzano; (567) ALPRRVKNLLLQVKLP(582), Catania; (580)KLP(582), Rugby Park; (572) LLQFSSF(578), and Venezia; (572)PTMRIRE(578)) were sub-cloned into the vector using the restriction sites *BamHI* and *XhoI* (GenScript, NJ, USA). The HSA variants were produced by transient transfection of adherent HEK293E cells (ATCC) using Polyethylenimine Max (Polysciences). To compare expression levels of WT HSA and the genetic variants, HEK293E cells were seeded in six-well plates and grown to 95–100% confluency before transfection. Growth medium was harvested and replaced on days 1, 2, 4, and 6 post transfection, and the levels of secreted protein were quantified using a two-way anti-HSA ELISA described below. For production of larger protein quantities, HEK293E cells were grown in T175 flasks and transfected at 95–100% confluency. Growth medium was harvested and replaced every second day for up to 2 weeks post transfection, before the HSA variants were purified from collected media using CaptureSelect HSA-affinity matrix (Life technologies) packed in a 5 ml column (Atoll). The column was pre-equilibrated with 1× PBS/0.05% sodium azide, before the supernatant was applied at a flow rate of 1–2 ml/min. The column was washed with 100 ml 1× PBS/0.05% sodium azide, before the bound protein was eluted with 50 ml 0.1 M Glycine-HCl (pH 3.0) and neutralized with 1 M Tris-HCl (pH 8.0). The collected protein was up-concentrated and buffer-exchanged to 1× PBS using Amicon Ultra-15 30 K columns (Millipore). Size-exclusion chromatography was performed using a Superdex 200 increase 10/300 GL column (GE Healthcare). The monomeric fraction was collected and up-concentrated using Amicon Ultra-0.5 30 K columns (Millipore). Protein concentrations were determined using a DS-11 spectrophotometer (DeNovix). Each HSA variant (2 μg) was mixed with LDS sample buffer (Novex) and distilled H$_2$O. The samples were loaded onto a 12% Bis-Tris NuPAGE SDS-PAGE gel in parallel with Spectra™ Multicolor Broad Range Protein Ladder (Thermo Fisher), and separated at 200 V for 22 min in MES buffer, followed by staining with Coomassie Brilliant Blue (Bio-Rad).

**Production of recombinant hFcRn.** A vector containing cDNA that encodes for the three ectodomains (α1–α3) of the hFcRn HC genetically fused to the *Schistosoma japonicum* glutathione *S*-transferase (GST), and the human β$_2$m subunit was used[64]. Soluble GST-tagged hFcRn was expressed by transient transfected HEK293E cells as described above, and purified using a GSTrap FF column (GE Healthcare). Briefly, the column was pre-equilibrated with 1× PBS/0.05% sodium azide, before the supernatant was applied at a flow rate of 1–2 ml/min. The column was washed with 100 ml 1× PBS/0.05% sodium azide, before bound receptor was eluted with 50 ml 10 mM reduced glutathione (Sigma-Aldrich) diluted in 50 mM Tris-HCl (pH 8.0). The collected receptor was up-concentrated and buffer-exchanged to 1× PBS using Amicon Ultra-15 10 K columns (Millipore).

For production of soluble His-tagged hFcRn, a Baculovirus expression vector system was used, as described previously[65]. The viral stock encoding His-tagged hFcRn was a kind gift from Dr. Sally Ward (University of Texas, Southwestern Medical Center, Dallas, TX). Briefly, the vector contains cDNA that encodes for the three ectodomains (α1–α3) of the hFcRn HC followed by six histidines, and the human β$_2$m subunit. High Five cells (Thermo Fisher Scientific) were cultured in Express FIVE SFM medium supplemented with 10 mM L-glutamine and 1% antibiotic-antimyotic (Thermo Fisher Scientific). The cells were cultured in suspension at a density of $1 \times 10^6$ cells/ml at 27 °C with shaking at 160 rpm, and on the day of infection, 1 ml of the viral stock was added per 500 ml of cell culture. Post infection, the cells were incubated at 23.5 °C with shaking at 160 rpm for 72 h, followed by centrifugation at 3500 rpm and 4 °C for 40 min. The supernatant was filtrated through a 0.2-μm filter unit (Millipore), and 0.05% sodium azide was added prior to storage at 4 °C. His-tagged hFcRn was purified using a HisTrap HP column (GE Healthcare) supplied with Ni$^{2+}$ ions as previously described[29]. Briefly, the supernatant was adjusted to pH 7.2 by adding 1× PBS/0.05% sodium azide (pH 10.9). The column was pre-equilibrated with 1× PBS/0.05% sodium azide, before the supernatant was applied at a flow rate of 5 ml/min. The column was washed with 150 ml 1× PBS/0.05% sodium azide and 50 ml 25 mM imidazole/1× PBS (pH 7.3), before bound receptor was eluted with 50 ml 250 mM imidazole/ 1× PBS (pH 7.4). The collected receptor was up-concentrated and buffer-exchanged to 1× PBS using Amicon Ultra-15 10 K columns (Millipore). Size-exclusion chromatography was performed using a HiLoad 26/600 Superdex 200 prep grade column (GE

Healthcare) coupled to an ÄKTA FPLC instrument (GE Healthcare). The monomeric fraction was collected and up-concentrated using Amicon Ultra-15 10 K columns (Milipore).

**CD spectroscopy.** CD spectra were recorded using a Jasco J-810 spectropolarimeter (Jasco International). Measurements were performed on WT HSA and L585X (150 μg/ml) in 10 mM PBS (pH 5.5) without NaCl at 20 °C using a quartz cuvette (Starna) with a path length of 0.1 cm. Each sample was scanned five times at 20 nm/min (bandwidth of 1 nm, response time of 1 s) with wavelength range set to 190–260 nm. The data were averaged, and the spectrum of a sample-free control was subtracted.

**hFcRn-HSA binding ELISA.** A human IgG1 mutant (M252Y/S254T/T256E/H433K/N434F) with specificity for 4-hydroxy-3-iodo-5-nitrophenylacetic acid[66] (8 μg/ml) diluted in PBS (pH 7.4) was added to 96-well plates (Costar) and incubated overnight at 4 °C. The wells were blocked with PBS containing 4% skimmed milk (PBSM) (pH 7.4) for 1 h at room temperature (RT), and then washed three times with PBS containing 0.005% Tween 20 (PBST) (pH 5.5). His-tagged hFcRn (10 μg/ml) diluted in PBSTM (pH 5.5) were added to the wells and incubated for 1 h at RT. HSA variants (15 μg/ml or twofold serial dilutions starting from 10 μg/ml) diluted in PBSTM (pH 5.5) was added to the wells in duplicates and incubated for 1 h at RT. Horseradish peroxidase-conjugated monoclonal mouse anti-HSA antibody (1:5000) (Abcam) or alkaline phosphatase-conjugated polyclonal goat anti-HSA antibody (1:3000) (Bethyl Laboratories, Inc.) diluted in PBSTM (pH 5.5) were added to the wells and incubated for 1 h at RT. The wells were washed three times with PBST (pH 5.5) after incubation of each layer. The wells were developed with tetramethylbenzidine substrate (Calbiochem) or p-nitropenyl-phosphate substrate (10 μg/ml) (Sigma-Aldrich) diluted in diethanolamine buffer. The absorbance was measured at 620 nm or 405 nm using the Sunrise spectrophotometer (TECAN).

**SPR.** SPR was performed using Biacore 3000 or T200 (GE Healthcare). Following the description provided by the manufacturer, CM5 sensor chips were coupled with GST-tagged hFcRn (~2000 RU) or HSA variants (~200–500 RU) using amine-coupling chemistry. Briefly, coupling was performed by injecting the receptor (10 μg/ml) or HSA (6–10 μg/ml) in 10 mM sodium acetate, pH 4.5 (GE Healthcare). For affinity and kinetic measurements, twofold serial dilutions of monomeric His-tagged hFcRn (0, 0.625, 1.25, 2.5, 5, 10, 20, 40, and 80 μM or 0, 0.0625, 0.125, 0.25, 0.5, 1, 2, 4, 8, 16, and 32 μM) were prepared in duplicates and injected over immobilized HSA with a flow rate of 10 or 40 μl/min at 25 °C. Relative binding of serum-derived HSA (WT HSA and L585X) was measured by injecting equal amounts (1 μM) over immobilized hFcRn at a flow rate of 50 μl/min and temperature of 25 °C. Phosphate buffer (67 mM phosphate buffer, 0.15 M NaCl, 0.05% Tween 20) at pH 5.5 or pH 7.4 was used as running and dilution buffer, while HBS-P buffer (0.01 M HEPES, 0.15 M NaCl, 0.005% surfactant P20) at pH 7.4 was used for regeneration of the flow cells. All binding curves were zero adjusted, and the reference cell value was subtracted. Binding kinetics and affinity values were estimated using the Langmuir 1:1 ligand binding model or an equilibrium (Req) binding model provided by the BIAevaluation 4.1 software or Biacore T200 Evaluation Software, version 3.0.

**HDX-MS.** All reagents were purchased from Sigma-Aldrich in analytical grade, except for the immobilized pepsin beads (Thermo Scientific). The exchange reaction was initiated by diluting HSA variants (20 pmol) 1:9 with 99% D$_2$O (111.2 mM Na$_2$HPO$_4$, 44 mM citric acid, pD$_{read}$ = 5.530) at 25 °C. After 15 s, 1 min, 10 min, 60 min, 480 min, and 2960 min, the exchange reaction was quenched by addition of ice-cold quench buffer (0.8 M TCEP in 2 M glycine, pH = 2.305) in a 1:1 relationship, thereby decreasing the pH$_{read}$ to 2.422. The quenched samples were immediately frozen to 80 °C until analysis. "Maximally-labeled" control samples were prepared by incubating samples for 2 days at 25 °C in presence of deuterated guanidine hydrochloride (6 M), where after the samples were treated as described above. All time points and the "maximally-labeled" samples were performed in triplicate except the 2960 min sample, which was only analyzed a single time.

Incorporation of deuterium was examined as previously described[67]. The quenched protein samples were defrosted and loaded onto a refrigerated reverse-phase UPLC-HDX system (Waters Inc.) cooled to 0 °C. The UPLC-HDX system was equipped with a self-packed pepsin column with an internal volume of 60 μl, which was kept externally to allow rapid online digestion of deuterated protein samples at 20 °C. The generated peptic peptides were captured on a C18 trap column (ACQUITY UPLC BEH C18 1.7 μm VanGuard column, Waters Inc.) and desalted for 3 min at 200 μL/min with 0.23% formic acid in water. After desalting, the flow path of the system was configured to allow the peptides to be eluted onto a C18 analytical column (ACQUITY UPLC BEH C18 1.7 μm, 1 × 100 mm column, Waters Inc.). Here, the peptic peptides were separated by short gradient (7 min) reversed-phase chromatography (solvent A, 0.23% formic acid in water; solvent B, 0.23% formic acid in acetonitrile) and ionized (into the gas-phase) by positive electrospray ionization into a hybrid Q-TOF mass spectrometer (Synapt G2-Si, Waters Inc.). Prior to mass analysis, peptide ions were separated according to their ion mobility. Identification of peptides was performed on non-deuterated and fully

reduced samples by tandem MS using both data-dependent and data-independent (MSe) acquisition methods.

Lock mass correction with Glu[1]-fibrinopeptide B was applied to all recorded spectra. To perform peptide identifications, the data were analyzed in PLGS 3.0, which correlate precursor- and fragment ions to a local database containing the sequence of WT HSA, pepsin, and randomized sequences of these two proteins as well. To determine the deuterium uptake for each peptide, the HDX-MS data were analyzed using the software DynamX 3.0 (Waters Inc.). To allow access to the HDX data of this study, the HDX Data Summary Table (Supplementary Table 2) and the HDX Data Table (Supplementary Data 1) are included in the Supporting Information according to the community-based recommendations[68].

**HERA**. In total, $7.5 \times 10^5$ HMEC1 cells stably expressing HA-hFcRn-EGFP[69] were seeded into 24-well plates per well (Costar) and cultured for 1 day in growth medium. The cells were washed twice and starved for 1 h in Hank's balanced salt solution (HBSS) (Life Technologies). WT HSA or L585X (1000 nM) diluted in 250 μl HBSS (pH 7.4) were added per well and incubated for 4 h. The medium was removed, and the cells were washed five times with ice-cold HBSS (pH 7.4). Growth medium supplemented with MEM nonessential amino acids (Thermo Fisher) was added and collected after 4 h. To quantify the amount of HSA in the samples, a two-way anti-HSA ELISA was used as described below.

**Mouse studies**. Two hFcRn transgenic mouse strains were used in the study: Hemizygous Tg32 mice (B6.Cg-Fcgrt $^{tm1Dcr}$ Tg(FCGRT)32Dcr/ DcrJ; The Jackson Laboratory, Bar Harbor, ME) and homozygous Tg32-Alb$^{-/-}$ albumin deficient mice (B6.Cg-Alb$^{em12Mvw}$ Fcgrt$^{tm1Dcr}$ Tg(FCGRT)32Dcr/Mvw; The Jackson Laboratory). Tg32-Alb$^{-/-}$ mice (male, 11–16 weeks, 20–25 g, four mice/group) received WT HSA or L585X (4 mg/kg) in 20 ml/kg 1× PBS by intraperitoneal injection, and blood (25 μl) was drawn from the retro-orbital sinus at 1, 8, 14, 20, 30, 45, and 58 days post injection. Tg32 mice (male, 7–8 weeks, 20–25 g, five mice/group) received WT HSA or L585X (2 mg/kg) in 10 ml/kg 1× PBS by intravenous injection, and blood (25 μl) was drawn from the retro-orbital sinus at 1, 2, 3, 5, 7, and 10 days post injection. The blood samples were mixed with 1 μl 1% K3-EDTA and maintained on ice until centrifugation at 17,000×$g$ for 5 min at 4 °C. Plasma was isolated and diluted 1:10 in 50% glycerol/PBS solution and stored at −20 °C. The animal studies were carried out at The Jackson Laboratory (Tg32 study by JAX Service, Bar Harbor, ME), and were performed in accordance with guidelines and regulations approved by the Animal Care and Use Committee at The Jackson Laboratory. To quantify the amount of HSA in plasma (diluted 1:200-1:400 in PBSTM), a two-way anti-HSA ELISA was used as described below. The plasma concentration was presented as percentage remaining in the circulation at different time points post injection compared to the concentration on day 1 (100%). The β-phase half-life was calculated using the formula: $t_{1/2} = \log 0.5/(\log A_e/A_0) \times t$, where $t_{1/2}$ is the half-life of the HSA variant evaluated, $A_e$ is the amount of HSA remaining, $A_0$ is the amount of HSA on day 1, and $t$ is the elapsed time.

**Two-way anti-HSA ELISA**. Polyclonal goat anti-HSA (1.0 μg/ml) (Sigma-Aldrich) or monoclonal mouse anti-HSA (1.0 μg/ml) (Abcam) diluted in PBS were added to 96-well ELISA plates (Costar) and incubated overnight at 4 °C. The wells were blocked with PBSM overnight at 4 °C, and then washed three times with PBST. Serial dilutions of HSA (500.0–0.2 ng/ml) in PBSTM were applied in parallel with collected growth medium, HERA samples or plasma, followed by 1 h incubation at RT. The wells were washed three times with PBST, before bound HSA was detected using alkaline phosphatase-conjugated polyclonal anti-HSA antibody from goat (1:3000) (Bethyl Laboratories, Inc.). The wells were washed three times with PBST and then developed with p-nitropenyl-phosphate substrate (10 μg/ml) (Sigma-Aldrich) diluted in diethanolamine buffer. The absorbance was measured at 405 nm using the Sunrise spectrophotometer (TECAN).

**ESI-MS and isolation of L585X from human plasma**. The patient included in the study experienced a single acute event of gallstone induced pancreatitis. Blood samples were taken upon hospital admission and then 2.2, 32.9, 35, 57.4, and 104.3 h after admission. Pancreatic amylase activity and lipase activity were measured on an Abbott c8000/c16000 analyzer (Abbott Laboratories) according to the manufacturer's protocol using a reference range of 8–53 U/L and 10–70 U/L, respectively. The total serum albumin concentrations were determined by bromocresol green binding method (normal range 35–50 g/L). To quantify the level of L585X in the blood, 10 μl of plasma was incubated for 2 h at 37 °C with 0.5 μl 150 mM dithiothreitol. Then, 1.5 μl of the mixture were separated by reverse-phase high-performance liquid chromatography on a Phemonenix C-4 column (25 × 0.46 cm). The HSA peak crest was collected, and 20 μl directly injected into the electrospray source of a Platform II mass spectrometer at a flow rate of 10 μl/min. Data were acquired and processed using MassLinks, and deconvoluted using Max-Ent software. The half-life of L585X was calculated using the formula: $t_{1/2} = \log 0.5/(\log A_e/A_0) \times t$, where $t_{1/2}$ is the half-life of L585X, $A_e$ is the percentage of L585X in the blood, $A_0$ is the percentage of L585X in the blood 2.2 h after hospital admission, and $t$ is the elapsed time since the 2.2 h blood sample.

HSA was isolated from a healthy control and from a patient with alcohol induced pancreatitis having 100% L585X, using DEAE Sephadex chromatography

employed 16 mM sodium acetate buffers with a pH gradient from pH 5.2 to 4.4[39]. The study was carried out in accordance with the Declaration of Helsinki principles and the Canterbury Health Laboratories guidelines on clinical samples. The patients gave written informed consent.

**CPA digestion**. HSA WT, K573P or Tanzeum® (15 μM) in 1× PBS were incubated with human pancreatic CPA (30 μg/ml) (Elastin Products Company, Inc.) for 5 h at 37 °C with shaking.

**LC–MS/MS**. LC–MS/MS was used to analyze HSA samples that had been incubated with or without CPA. Each sample (3 μg) was mixed with 50 mM NH$_4$HCO$_3$ (pH 7.8) to a volume of 18 μl. For protein reduction, 2 μl 100 mM dithiothreitol was added and incubated for 30 min at 56 °C. For alkylation, 5 μl 55 mM iodoacetamide was added and incubated in the dark for 30 min at RT. For protein digestion, trypsin (0.5 μg) was added to each sample and incubated in a wet chamber overnight at 37 °C. The samples were subsequently purified on C18 micro columns: three discs were assembled in a 200 μl pipette tip and activated by 50 μl MeOH followed by 50 μl ACN. The discs were equilibrated twice by 80 μl 0.1% TFA before the samples were added. The discs were washed twice by 80 μl 0.1% FA before the samples were eluted by 80 μl 80% ACN/0.1% FA. Then, the samples were centrifuged in a speed vacuum concentrator to remove ACN, and the volume was adjusted to 7 μl using 0.1% FA. Thus, the samples were diluted seven times, and then stored at −20 °C until analysis. The samples were analyzed using an EASY-LC coupled to Q Exactive Quadrupole-Orbitrap mass spectrometer (Thermo Fisher) with the EASY Spray PepMap®RSLC column (C18, 2 μl, 100 Å, 75 μm × 25 cm) and column temperature at 60 °C. A 60 min gradient and injections of 3 μl were used.

MS raw files were submitted to MaxQuant software (version 1.6.1.0) for protein identification and rough quantification. Searches were done against the sequences of WT HSA and L585X. Parameters were set as follow: carbamidomethylation as fixed modification, and protein N-acetylation and methionine oxidation as variable modifications. First search error window of 20 ppm and mains search error of 4.5 ppm. Trypsin with two allowed miscleavages was used. Quantification was done on the peptide level by comparing the intensity of the C-terminal peptide lacking L585 against the intensity of the corresponding WT peptide within the same sample, without correlating for possible differences in ionization efficiency of the single peptides.

**Sequence and structural analysis**. The albumin sequences of human (AAA98797), orangutan (NP_001127106.2), chimpanzee (XP_517233.3), macaque (NP_001182578.1), panda (XM_002928492.3), elephant (AAT90502.1), horse (NP_001075972.1), donkey (AAV28861), cattle (AAA51411), goat (ACF10391), sheep (NP_001009376), pig (AAA30988.1), dog (CAB64867.1), cat (CAA59279.1), rabbit (NP_001075813), opossum (XM_001364821.2), hamster (ABR68005.1), guinea pig (AAQ20088.1), rat (AAH85359.1), and mouse (AAH49971) were downloaded from the National Center for Biotechnology Information. The alignments were made using Clustalω software.

The coordinates of the crystal structure of HSA (PDB ID 1AO6)[3] and of HSA in complex with hFcRn (PDB ID 4N0F)[26] were used and inspected using the PyMOL software (Schrodinger Inc.).

**Statistics and reproducibility**. Statistical analysis of the data obtained by HDX-MS was performed using Excel software (Microsoft). HDX was performed with triplicates of each albumin variant for all time points, except for the 2960 min sample where $n = 1$. For data points performed in triplicate, the comparative analyses were performed with either a homoscedastic or a heteroscedastic Student's $t$ test, depending on the equality of the variances of the compared data points. The equality of the variances was determined by an $F$ test with the significance level set to 0.05. A peptide was only considered to have a significant difference in HDX if one of its data points met the following two criteria[70]: (1) a significant difference in deuterium uptake ($p < 0.05$), and (2) the absolute difference in deuterium uptake should be larger than two times the pooled standard deviation ($|\Delta D| > 2 \times \sqrt{SD_{wt}^2 + SD_{L585X}^2} = 0.13D$) (Supplementary Fig. 7). For the last (2960 min) time point, for which replicate data were not obtained, statistical significance was determined if the change exceeded a 99% confidence limit (±0.26 D) following the method described by Houde et al. and Arora et al.[71,72]. This 99% confidence limit was estimated from 385 single measurements of standard deviations from each protein state.

Statistical analysis of the data obtained by HERA and half-life measurements in mice was performed using GraphPad Prism 8 (GraphPad Software Inc.). The HERA experiment was performed three times, where three wells for each albumin variant was included each time. Half-life measurements were performed once in Tg32-Alb$^{-/-}$ mice and once in Tg32 mice, where four and five mice per albumin variant were included, respectively. Statistical significance was assessed by an unpaired Student's $t$ test (with 95% confidence level and a two-sided $p < 0.05$ defined as a significant difference).

**Reporting summary**. Further information on research design is available in the Nature Research Reporting Summary linked to this article.

## Data availability

Data that support the findings of this study are stored in excel file format and are available from the corresponding author upon reasonable request. Source data underlying graphs presented in the main figures are available in Supplementary Data 1 and 2.

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

## Acknowledgements

We are grateful to Sathiaruby Sivaganesh for excellent technical assistance. We thank Dr. Wayne I. Lencer for (Boston Children's Hospital, Harvard Medical School and Harvard Digestive Diseases center) for the HMEC1 cell line stably expressing HA-hFcRn-EGFP. This work was supported in part by the Research Council of Norway through its Center of Excellence funding scheme (project no. 179573). J.T.A. and J.N. were supported by the Research Council of Norway (Grant no. 230526) and South-Eastern Norway Regional Health Authority (Grant no. 40018). E.T. and K.D.R. were supported by the Benzon Foundation and the Independent Research Foundation Denmark (Sapere Aude Grant-DFF-4184-00537A). A.G. and K.M.K.S. were supported by the University of Oslo. C.A. was supported by Fundação para a Ciência e a Tecnologia (FCT) Portugal (SFRH/BD/117598/2016). M.B. was supported by the Research Council of Norway through its program for Global Health and Vaccination Research (GLOBVAC) (Grant no. 143822) and program for biotechnology and innovation (BIOTEK2021) (Grant no. 267606). B.D. was supported by South-Eastern Norway Regional Health Authority (Grant no. 2015095) (Regional Core Facility for Structural Biology).

## Author contributions

J.N., E.T., A.G., S.O.B., M.S., B.D., D.C.R., I.S., K.D.R., and J.T.A. designed research; J.N., E.T., A.G., C.A., S.O.B., M.S., J.W., K.M.K.S., M.B., and J.T.A. performed research; J.N., E.T., A.G., S.O.B., M.S., B.D., D.C.R., I.S., and J.T.A. analyzed the data; J.N., E.T., I.S., and J.T.A. wrote the paper.

## Competing interests

The authors declare no competing interests.
