## [Peer Review File · Communications Biology]

Reviewers' comments:

Reviewer #1 (Remarks to the Author):

The manuscript by Nilsen and colleagues describes the effect of removal of the last C-terminal residue (and other C-terminal truncations) of albumin on binding to FcRn and pharmacokinetic behavior. The study also involves an analysis of the generation of C-terminally truncated albumin that occurs in humans during acute pancreatitis. Using HDX-MS analysis, the authors show that lack of the C-terminal residue, L585, results in structural stabilization in a flexible region of albumin that is involved in binding to FcRn.

The studies reveal novel aspects of albumin function that have relevance to understanding both human disease and the use of (engineered) albumin to extend the half-life of therapeutics. The manuscript is clearly written and the data support the conclusions. I have the following specific comments:

1. Page 8, lines 182-184. The mouse group size is small for these analyses, especially since only one experiment is carried out. Is the difference in half-life (14 vs. 18 hours) significant? From the data shown in Table 2, it looks as if it might not be. Can the authors comment on this?
2. Page 9, lines 208-210. Although stated in the methods, it is not clear in the results how plasma-derived L585X was purified. It would be helpful to state that the source was a patient with 100% L585X.
3. Fig. S2. Presumably the symbols in panel A are overlapping? Can these data be more clearly displayed?
4. Fig. S3. The font sizes on these panels are very small and hard to read. Although space is limited and the panel number is very large, could these panels be made to look more like those shown in Figure 3?

Minor comment

1. Line 473. Presumably 'speed aced' is a typographical error? There are also some other minor grammatical errors in the methods section.

Reviewer #2 (Remarks to the Author):

The authors researched how structural alterations impacted albumin transport and pharmacokinetics. They reported that the integrity of C-terminal end, particularly the L585 residue, is crucial for receptor binding and subsequent recycling. A series of techniques were employed to study the interaction between albumin (and its variants) and its receptor hFcRn, including circular dichroism (CD) spectroscopy, ELISA, human endothelial cell-based recycling assay (HERA), hydrogen/deuterium exchange mass spectrometry etc. Pharmacokinetics was investigated in both mouse and human samples.

In general, I found the manuscript is appealing to the readers of Communications Biology, in particular those working on albumin-based therapeutics. Adequate experimental and computational evidences are provided in a logical manner to support the claims. Following I have a few minor points to be further addressed.

1. The full name of SPR is not given anywhere.
2. Figure 4B&C, the title of Y-axis "HSA remaining" is a bit confusing. I guess it is the relative HSA concentration, rather than amount.

3. Figure 5B, did the authors measure absolute concentration of total HSA? If yes, are there any rationales for that the relative rather than absolute concentration of L585X was shown? The half-life of L585X (Table 2, Page 9 Line 208) was calculated based on its relative concentration. I think this is inappropriate unless the concentration of total HSA was constant.
4. Figure S5, how the two peaks were assigned to full-length HSA and L585X? Neither the detected m/z nor the difference in m/z of the peaks matches with theoretical value(s).

Reviewer #3 (Remarks to the Author):

This paper is reported the new strategy for the design of albumin therapeutics with tailored pharmacokinetics. Authors carefully designed the experimental protocol. They provides interesting findings and the results obtained here will present useful and valueable data for the new albumin drug developments including albumin DDS developments. I enjoyed it very much for this paper.

Reviewer #4 (Remarks to the Author):

Review of Manuscript Number: COMMSBIO-19-1354

Corresponding Author: Jan Terje Andersen

Title: An intact C-terminal end of albumin is required for long half-life

Summary:

This manuscript by Nilsen et. al. investigated in depth the interaction between albumin and FcRn and demonstrates that albumin variants have reduced binding to FcRn, explaining their shorter plasma half-life. The authors conclude that this helps understand albumin homeostasis and design of albumin therapeutics with long half-lives, an area of immediate relevance for folks in the field, and the scientific contributions are considerable. I recommend publication providing the two major points I list below are addressed to remove technical flaws.

Major comments:

- Results lines 110-116, Figure 1D-H, 2E and Table 1: Contain SPR data measuring affinity to FcRn to albumin variants. I agree that Biacore affinity measurement of kinetic KD (the KD derived from the ratio of the off and on rates) will match the KD derived from a steady-state experiment. A well-designed experiment includes (1) concentration ranges spanning the KD (0.1 X KD to 10 X KD), (2) clear signs of saturation of the binding sites at the highest concentration(s) and (3) the data fitting well to the 1:1 model applied. The reported concentrations of FcRn injected are 80-0.0625 uM. For determination of steady-state KD, only concentration ranges spanning the KD (0.1 X KD to 10 X KD) should be used. It is especially important to not include concentrations lower than 0.1 X KD as those will skew the fit more significantly. The steady-state data should be reanalyzed, using only the concentrations in this range to report steady-state affinity values (for example, fit the ranges from either 0.625-80 or 1.25-80 for the variants). This will adjust the uM binding affinities reported in Table 1, but not alter your conclusion that the affinity of the HSA variants is lower than that of the WT. In the figures and method sections, report the concentrations injected, not just the range, this is preferred and more transparent. Once the KD values are adjusted the methods section should be edited appropriately (it seems you used 2-fold dilutions for the HSA variants at 0.156, 0.313, 0.625, 1.25, 2.5, 5, 10, 20, 40 and 80 nM concentrations; stating that would be transparent for those wanting to reproduce data). Figures 1D and 2E should include the data fit to the 1:1 model, so readers can see how well the fit matches the sensorgrams.
- Results lines 154-170, Figure 3B, 3C, 3D, 3E: contain the HDX-MS analysis. It is claimed that

Figures 3B-D show significant difference in deuterium incorporation (are there statistics that confirm this statement?). I only see minor differences between the variants in Figure 3D and not much of a difference in 3B and 3C, those are subtle at best. Examining Figure S3 and looking for peptides that overlap with those in 3B (409-420), it seems that 2 days is not long enough to reach saturation for those peptides. The data and manner in that it is reported is difficult to interpret without statistical values to support the conclusions.

Minor comments:

- Results line 98: the first Figure is 1I, not 1A. This is unconventional.
- Results lines 109-110 and Figure 1B/supplement figure S1: More description is needed in the text or figure legend to clarify. In the lines in the text the authors comment that only minor expression levels are seen in comparison to WT albumin, Figure 1C shows relative expression levels; one assumes the levels have been normalized to WT albumin, but that is not stated. What is measured to determine expression? Capture from conditioned medium or final purified material? In Figure 1B, the final purified protein fraction was loaded, again, one assumes a standard amount was loaded per well, this is not stated. Was any additional characterization done, % monomeric? One can see higher bands, running between the 70-100kD markers in the Rugby Park, Venezia, Catania and Bazzano bands.
- Results Figure 1D and 2E: The fit to the 1:1 model should be visible here.
- Results lines 134-135: state directly and reference that goat IgG does not bind human FcRn. If no reference then include a control in Figure S2A showing the anti-HSA goat IgG secondary only with no HSA added.
- Results line 192-194: It might also be prudent to report here (or earlier) the differences in kinetics of WT vs L585X. The L585X has a slower on rate and a faster off rate compared to WT. In the endosome it may be less likely to bind FcRn because of its kinetics; this point can be reiterated in the discussion.
- Results lines 208-210: Is this plasma from a normal subject? Pooled serum from healthy donors? Explain. You mentioned earlier that 4-15% of HSA is L585X, could you quantitate the amount of L585X HSA from this normal HSA? Is this the data in Table S1?
- Results lines 221-222: Restate the relevance of the HSA K573P mutant here; it was mentioned at the end of the introduction, but those not familiar with HSA mutations/Veltis technology will have forgotten.
- Discussion lines 264-272: It is mentioned that the HSA C-term leucine is conserved in human and monkey and the human FcRn binding, was monkey FcRn binding measured previously with WT and L585X? What about binding to other species of FcRn (i.e. mouse or rat)? Is there no difference there because L585 is not present in those species of HSA? This also applies to lines 286-288.
- Discussion lines 273-275: Report the level of normal HSA that lacks L585 in your experiments here.
- Discussion lines 296-298: The alternative point - C-term fusion of a drug to HSA might not be favorable if enzymatic cleavage occurs when the drug is fused.
- Discussion lines 301-304: Has this K573P variant been in the clinic? Is there any evidence of immunogenicity?

Point by point responses to reviewers' comments

Reviewer #1 (Remarks to the Author):

The manuscript by Nilsen and colleagues describes the effect of removal of the last C-terminal residue (and other C-terminal truncations) of albumin on binding to FcRn and pharmacokinetic behavior. The study also involves an analysis of the generation of C-terminally truncated albumin that occurs in humans during acute pancreatitis. Using HDX-MS analysis, the authors show that lack of the C-terminal residue, L585, results in structural stabilization in a flexible region of albumin that is involved in binding to FcRn.

The studies reveal novel aspects of albumin function that have relevance to understanding both human disease and the use of (engineered) albumin to extend the half-life of therapeutics. The manuscript is clearly written and the data support the conclusions. I have the following specific comments:

1. Page 8, lines 182-184. The mouse group size is small for these analyses, especially since only one experiment is carried out. Is the difference in half-life (14 vs. 18 hours) significant? From the data shown in Table 2, it looks as if it might not be. Can the authors comment on this?

We thank the reviewer for commenting on this and giving us the opportunity to perform statistical analysis on the half-life data. Importantly, the mean β -phase half-lives in table 2 are given in days. Thus, a half-life of 14 days was measured for L585X, while 18 days was measured for the WT in the mouse model lacking albumin (Tg32-Alb^{-/-}). We performed an unpaired Student's t-test to test for difference in mean half-life, which showed a significant difference ($p < 0.05$) between the WT and L585X (18 vs. 14 days). The same test was performed with the half-life data from the mouse model expressing albumin (Tg32), which also demonstrated a significant difference ($p < 0.005$) between the WT and L585X (2.7 vs. 1.6 days). The statistical analysis was added to the method section (page 22, lines 520-522), figure 4 and the figure legend (page 30, lines 783-784). The half-life values were moved from table 2 (which have been deleted) to figure 4.

2. Page 9, lines 208-210. Although stated in the methods, it is not clear in the results how plasma-derived L585X was purified. It would be helpful to state that the source was a patient with 100% L585X.

We appreciate this comment from the reviewer, and have now added the following statement to the results part: *"We isolated truncated HSA from a patient with 100% L585X, while full-length HSA was isolated from a healthy individual in which L585X accounted for about 3% of the total albumin"*. (Page 9, lines 208-210).

3. Fig. S2. Presumably the symbols in panel A are overlapping? Can these data be more clearly displayed?

We thank the reviewer for making us aware of this. The symbol representing L585X has now been changed to triangle and can be distinguished from the open circles representing WT HSA (Supplementary Information, page 3, Supplementary Fig 2).

4. Fig. S3. The font sizes on these panels are very small and hard to read. Although space is limited and the panel number is very large, could these panels be made to look more like those shown in Figure 3?

We thank the reviewer for addressing the readability of the plots. To meet the request of the reviewer, we have supplied these Supporting information plots in a very high resolution (vector format), thus a reader can zoom at least up to 10 times allowing all details of the plots to be seen clearly. We are

convinced that this will allow the reader to easily access and assess all HDX data described in the main text of the manuscript.

Minor comment

1. Line 473. Presumably ‘speed aced’ is a typographical error? There are also some other minor grammatical errors in the methods section.

We thank the reviewer for recognizing the typo. We have corrected “speed aced” to “speed vaced” (Page 20, line 482), as well as other grammatical errors we could identify in the methods section.

Reviewer #2 (Remarks to the Author):

The authors researched how structural alterations impacted albumin transport and pharmacokinetics. They reported that the integrity of C-terminal end, particularly the L585 residue, is crucial for receptor binding and subsequent recycling. A series of techniques were employed to study the interaction between albumin (and its variants) and its receptor hFcRn, including circular dichroism (CD) spectroscopy, ELISA, human endothelial cell-based recycling assay (HERA), hydrogen/deuterium exchange mass spectrometry etc. Pharmacokinetics was investigated in both mouse and human samples.

In general, I found the manuscript is appealing to the readers of Communications Biology, in particular those working on albumin-based therapeutics. Adequate experimental and computational evidences are provided in a logical manner to support the claims. Following I have a few minor points to be further addressed.

1. The full name of SPR is not given anywhere.

We thank the reviewer for this comment, and have now written the full name when it was first mentioned (Page 5, lines 107-108).

2. Figure 4B&C, the title of Y-axis “HSA remaining” is a bit confusing. I guess it is the relative HSA concentration, rather than amount.

We thank the reviewer for the comment and can understand that the label may be confusing. Indeed, the serum concentrations of the albumin variants (WT and L585X) are presented as the percentage remaining in the circulation compared to that measured 24 h after injection (day 1 = 100%). To avoid any confusion, the title of the Y-axis was changed to “percent remaining” (Figure 4 b and c).

3. Figure 5B, did the authors measure absolute concentration of total HSA? If yes, are there any rationales for that the relative rather than absolute concentration of L585X was shown?

The half-life of L585X (Table 2, Page 9 Line 208) was calculated based on its relative concentration. I think this is inappropriate unless the concentration of total HSA was constant.

We thank the reviewer for these questions and giving us the opportunity to clarify. The percentage of L585X of the total albumin in Figure 5b is based on the mass data in Supplementary Figure 5, which shows the actual output i.e. the relative abundance of each albumin isoform at each time point. We have altered the y-axis to clarify this and indicate the most abundant peak is always at 100% full scale (Supplementary Figure 5b). We also determined the total concentration of albumin in each of the six blood samples taken from the pancreatitis patient by bromocresol green binding method. These data have been added to Supplementary Figure 5a. The result (page 9, lines 202-204) and method section (page 19, lines 455-456) were updated accordingly. The total albumin concentration falls slightly over the 104 h time period (from 40.9 g/L at 0 h to 35.9 g/L at 104 h), which is expected as albumin is a negative acute phase protein, meaning that the liver downregulates synthesis during inflammation. In addition, the generation of high levels of L585X, which we now show has reduced ability to bind and be recycled by FcRn, is likely to contribute to the reduction in total albumin concentration. We hope

that the reviewer agree that, in this type of *in situ* patient study, one cannot expect the total albumin concentration to be constant.

4. Figure S5, how the two peaks were assigned to full-length HSA and L585X? Neither the detected m/z nor the difference in m/z of the peaks matches with theoretical value(s).

Supplementary Figure 5b shows a Deconvoluted (transformed) ESI-MS spectra and not an m/z spectrum. The assignment of peaks as full-length HSA or L585X was performed as described in our previous publications (Ref. 39 and 53; Brennan et al., 1999 and 2000), where we employed mass mapping of CNBr fragments and tryptic peptides together with carboxypeptidase A digestion. The average mass difference between the major albumin components is 110 Da, which is close to the expected difference (113 Da) for loss of Leu585 and well within the expected error of the mass spectrometer used.

Reviewer #3 (Remarks to the Author):

This paper is reported the new strategy for the design of albumin therapeutics with tailored pharmacokinetics. Authors carefully designed the experimental protocol. They provides interesting findings and the results obtained here will present useful and valueable data for the new albumin drug developments including albumin DDS developments. I enjoyed it very much for this paper.

Reviewer #4 (Remarks to the Author):

Review of Manuscript Number: COMMSBIO-19-1354

Corresponding Author: Jan Terje Andersen

Title: An intact C-terminal end of albumin is required for long half-life

Summary:

This manuscript by Nilsen et. al. investigated in depth the interaction between albumin and FcRn and demonstrates that albumin variants have reduced binding to FcRn, explaining their shorter plasma half-life. The authors conclude that this helps understand albumin homeostasis and design of albumin therapeutics with long half-lives, an area of immediate relevance for folks in the field, and the scientific contributions are considerable. I recommend publication providing the two major points I list below are addressed to remove technical flaws.

Major comments:

• Results lines 110-116, Figure 1D-H, 2E and Table 1: Contain SPR data measuring affinity to FcRn to albumin variants. I agree that Biacore affinity measurement of kinetic KD (the KD derived from the ratio of the off and on rates) will match the KD derived from a steady-state experiment. A well-designed experiment includes (1) concentration ranges spanning the KD (0.1 X KD to 10 X KD), (2) clear signs of saturation of the binding sites at the highest concentration(s) and (3) the data fitting well to the 1:1 model applied.

The reported concentrations of FcRn injected are 80-0.0625 uM. For determination of steady-state KD, only concentration ranges spanning the KD (0.1 X KD to 10 X KD) should be used. It is especially important to not include concentrations lower than 0.1 X KD as those will skew the fit more significantly. The steady-state data should be reanalyzed, using only the concentrations in this range to report steady-state affinity values (for example, fit the ranges from either 0.625-80 or 1.25-80 for the variants). This will adjust the uM binding affinities reported in Table 1, but not alter your conclusion that the affinity of the HSA variants is lower than that of the WT.

The comment from the reviewer is appreciated. We are agree that it is important to use concentration ranges spanning the KD by 0.1-10 fold, and have re-analyzed the SPR data applying concentrations in

the 0.625-80 μ M (Catania and Venezia) or 1.25-80 μ M (Bazzano and Rugby Park) range to determine the steady-state affinity values. This altered the affinity values slightly but, as the reviewer pointed out, did not alter the conclusions drawn from the data. Figure 1 and Table 1 have been updated accordingly.

In the figures and method sections, report the concentrations injected, not just the range, this is preferred and more transparent. Once the KD values are adjusted the methods section should be edited appropriately (it seems you used 2-fold dilutions for the HSA variants at 0.156, 0.313, 0.625, 1.25, 2.5, 5, 10, 20, 40 and 80 nM concentrations; stating that would be transparent for those wanting to reproduce data).

We thank the reviewer for the suggestion and agree that including each concentration will make the experimental set-up more transparent. We have now stated each concentration injected in the method section (Page 15, lines 368-369), in figure legend 1 (Page 28, lines 737-738 and lines 740-741) and figure legend 2 (Page 28-29, lines 754-755).

Figures 1D and 2E should include the data fit to the 1:1 model, so readers can see how well the fit matches the sensorgrams.

We thank the reviewer for this comment. The fit of the data to the 1:1 binding model is now shown in Figure 1D and 2E. The figure legends were updated accordingly (Page 28, lines 739 and page 28, lines 755-756).

• Results lines 154-170, Figure 3B, 3C, 3D, 3E: contain the HDX-MS analysis. It is claimed that Figures 3B-D show significant difference in deuterium incorporation (are there statistics that confirm this statement?). I only see minor differences between the variants in Figure 3D and not much of a difference in 3B and 3C, those are subtle at best. Examining Figure S3 and looking for peptides that overlap with those in 3B (409-420), it seems that 2 days is not long enough to reach saturation for those peptides. The data and manner in that it is reported is difficult to interpret without statistical values to support the conclusions.

We agree with the reviewer that sound statistical analysis should be performed of all HDX-MS data before it can support claims on changes in protein dynamics. For this reason, we would like to draw the attention of the reviewer to the “Materials and Methods” section (subsection “Statistical Analysis”, Page 21, lines 507-519), where we describe the statistical approach for analyzing all our HDX-MS data. All cases described in the text as “significant differences in HDX” (see for instance the legend of Fig. 3) were indeed significant according to this statistical analysis procedure. The statistical analysis is a combination of two already published methods by Bennett et al, Houde et al. and Arora et al., depending on the number of replicates of the investigated time points. Our observations of significant changes in HDX (Fig. 3b, 3c, 3d) were further supported by the presence of several overlapping peptides that show similar significant changes in HDX, as also described in the current version of the manuscript on Line 164-166 (Page 7):

“The significance of these reductions in HDX was further supported in data from several overlapping peptides comprising these regions of DIII (Supplementary Fig. 3).”

The actual overlapping peptides referred to as showing a significant change in HDX are:

Region 1: peptides 408-420, 409-420 and 409-423

Region 2: peptides 492-506, 493-506 and 496-506

Region 3: peptides 530-544, 530-546, 530-548, 530-550 and 531-546.

For reviewer convenience we have included the relevant subsection of the “Materials and Methods” on statistical analyses below (now with minor changes made to help readability, line 507-519) (Page 21):

“Statistical analyses of the data obtained by HDX-MS was performed using Excel software (Microsoft). For data points performed in triplicate, the comparative analyses were performed with either a homoscedastic or a heteroscedastic Student’s t-test, depending on the equality of the variances of the compared data points. The equality of the variances was determined by an F-test with the significance level set to 0.05. A peptide was only considered to have a significant difference in HDX if one of its data points met the following two criteria (70): 1) A significant difference in deuterium incorporation ($p < 0.05$) and 2) the absolute difference in deuterium uptake should be larger than two times the pooled standard deviation ($|\Delta D| > 2 \times \sqrt{SD_{wt}^2 + SD_{L585X}^2} = 0.13D$). For the last (49 h and 20 min) time point, for which replicate data was not obtained, statistical significance was determined if the change exceeded a 99 % confidence limit ($\pm 0.26 D$) following the method described by Houde et. al. and Arora et. al (71,72). This 99 % confidence limit was estimated from 385 single measurements of standard deviations from each protein state.”

As correctly pointed out by the reviewer, incubation of the samples in D₂O for two days is not enough time to reach full deuteration of several peptides. This is a common feature in HDX-MS experiments, as the exchange half-life for very protected backbone amide hydrogens can be as long as weeks and months. The maximally labelled sample shows that approximately half of the peptide 409-420 has still not exchange in the probed time window. We can hypothesize, that the difference in exchange would also be significant at even longer time points than 2 days, but from the obtained data, we cannot not confirm this. However, this does not disregard the observed differences in exchange present already after 49h and 20min.

To accommodate the request of the reviewer, we have now added **the volcano plot associated with our statistical analysis of all data available in replicates** (Supplementary Figure 7, and see below). We have furthermore marked all data points that show significant differences in HDX according to our statistical analyses in the HDX plots of Figure 3. The figure legend has been updated accordingly (Page 29, lines 764-765).

Supplementary Figure 7. Scatterplot of replicate HDX data of the comparison of WT HSA and L585X. The difference in HDX is plotted on the y-axis, while 1 subtracted the p-value of the result of a student’s t-test is plotted on the x-axis. Each point represents a single peptide at a single time point. The horizontal dashed lines represents a two times the pooled standard deviation ($\pm 2 \times \sqrt{SD_{wt}^2 + SD_{L585X}^2} = \pm 0.13D$) cutoff based on the noise in the HDX measurements. The vertical dashed line represents a 1-p cutoff value of 0.95.

Minor comments:

- **Results line 98: the first Figure is 1I, not 1A. This is unconventional.**

We agree, and have removed the reference to the figure as it is not needed there.

- **Results lines 109-110 and Figure 1B/supplement figure S1: More description is needed in the text or figure legend to clarify.**

In the lines in the text the authors comment that only minor expression levels are seen in comparison to WT albumin, Figure 1C shows relative expression levels; one assumes the levels have been normalized to WT albumin, but that is not stated. What is measured to determine expression? Capture from conditioned medium or final purified material?

We thank the reviewer for making us aware of the limited information provided on how the expression levels of WT HSA and the genetic variants were measured and compared. To compare expression levels of WT HSA and the genetic variants, HEK293E cells were seeded in 6-well plates and grown to 95-100% confluency before transfection. Growth medium was harvested and replaced on day 1, 2, 4 and 6 post transfection, and pooled before the levels of secreted protein were quantified using a two-way anti-HSA ELISA. This description has been added to the method section (Page 13-14, lines 321-327). The use of ELISA to quantify the level of secreted protein in conditioned medium has also been stated in the legend of Figure 1c (Page 28, lines 735-736). The expression level of each of the genetic HSA variants were normalized to that of WT HSA, which was set to 1 (WT=1). This has now shown on the Y-axis of Figure 1c.

In Figure 1B, the final purified protein fraction was loaded, again, one assumes a standard amount was loaded per well, this is not stated. Was any additional characterization done, % monomeric? One can see higher bands, running between the 70-100kD markers in the Rugby Park, Venezia, Catania and Bazzano bands.

Indeed, 2 μ g of purified protein was loaded of each HSA variant onto the 12% SDS-PAGE gel. This has now been stated in the legend of Figure 1b (and Figure 2b) (Page 28, lines 733 and 751). In addition, a description of how the SDS-PAGE analysis was conducted has been added to the method section (Page 14, lines 336-340).

The variants were purified from conditioned medium using a Capture select affinity column specific for human albumin, and subsequently gelfiltrated using a Superdex 200 increase 10/300 GL column (size exclusion chromatography). As pointed out by the reviewer, a weak higher band between the 70-100 kDa markers is present in addition to the main band at about 66.5 kDa. This minor fraction of higher molecular weight was also visible on the size exclusion chromatography profile, in which the peak absorbance was only 5% compared to the main fraction.

- **Results Figure 1D and 2E: The fit to the 1:1 model should be visible here.**

We thank the reviewer for this comment. The fit of the data to the 1:1 binding model is now shown in Figure 1d and 2e. The figure legends were updated accordingly (Page 28, lines 739 and page 29, lines 755-756).

- **Results lines 134-135: state directly and reference that goat IgG does not bind human FcRn. If no reference then include a control in Figure S2A showing the anti-HSA goat IgG secondary only with no HSA added.**

We agree with the reviewer that the possibility for the goat IgG antibody to bind human FcRn is important to keep in mind. In this case, the ELISA design ensures that the detection antibody does not

bind via its Fc, since the IgG binding site on FcRn is already occupied (bound strongly to the Fc engineered antibody in the coat) when it is added. We have further clarified this in the text (Page 6, lines 132-133):

“Importantly, the detection antibody cannot bind via its Fc to FcRn, as the IgG binding site is already occupied when added.”

A control close to that suggested by the reviewer is included as part of the experiment shown in Supplementary Fig. 6. In this experiment, the ELISA set-up was used to examine hFcRn binding of albumin variants before and after exposure to carboxypeptidase A. The concentration of the albumin variants were titrated, while the amount of the detection antibody was constant. Importantly, no signal was detected when albumin was added at low concentrations, supporting that the detection antibody does not bind to the receptor in this set-up.

• Results line 192-194: It might also be prudent to report here (or earlier) the differences in kinetics of WT vs L585X. The L585X has a slower on rate and a faster off rate compared to WT. In the endosome it may be less likely to bind FcRn because of its kinetics; this point can be reiterated in the discussion.

We appreciate this suggestion from the reviewer, and have now discussed the cellular and in vivo behavior of L585X in light of its altered binding kinetics (Page 8, lines 190-193):

“HSA lacking L585, which binds hFcRn with 5-fold weaker affinity due to a slower on-rate and a faster off-rate, will less likely bind to the receptor in the endosomes and especially in the presence of large amounts of competing endogenous albumin, and instead follow the default pathway to the lysosomes.”

• Results lines 208-210: Is this plasma from a normal subject? Pooled serum from healthy donors? Explain. You mentioned earlier that 4-15% of HSA is L585X, could you quantitate the amount of L585X HSA from this normal HSA? Is this the data in Table S1?

We thank the reviewer for these questions. Full-length HSA was isolated from plasma taken from a healthy individual in which L585X accounted for about 3% of the total albumin. The Deconvoluted (transformed) ESI-MS spectrum of HSA in plasma collected from the healthy control was added to Supplementary Figure 5c, and a sentence was added to the result section to clarify (Page 9, lines 208-210).

The data in Table S1 relates to another experiment in which recombinant HSA variants were incubated with or without carboxypeptidase A, followed by MS analysis to determine the percentage lacking L585. As indicated in the table, more than 99% of the recombinant HSA incubated with carboxypeptidase A were consequently lacking the leucine, while for recombinant HSA incubated without the enzyme, this was the case for 1.2% or less.

• Results lines 221-222: Restate the relevance of the HSA K573P mutant here; it was mentioned at the end of the introduction, but those not familiar with HSA mutations/Veltis technology will have forgotten.

We agree with the reviewer and have restated relevant information about the HSA K573P variant in the results part (Page 9-10, lines 223-226):

“In addition, we included the engineered variant, HSA K573P, which has improved pH dependent binding affinity for hFcRn and extended half-life in non-human primates. Incubation with CPA resulted in complete removal of L585 from the C-terminal end of the engineered variant (Supplementary Table 1).”

• Discussion lines 264-272: It is mentioned that the HSA C-term leucine is conserved in human and monkey and the human FcRn binding, was monkey FcRn binding measured previously with WT and L585X? What about binding to other species of FcRn (i.e. mouse or rat)? Is there no difference there because L585 is not present in those species of HSA? This also applies to lines 286-288.

We agree that investigating how albumin binds FcRn across species is an important aspect, especially since rodents and monkeys often are used as preclinical animal models to evaluate HSA-based therapeutics. We have previously reported that large differences indeed exist, including between the mouse and human proteins, in which case HSA binds poorly to mouse FcRn, while mouse albumin binds strongly to hFcRn (Andersen et al., 2010, 2012, 2014). Moreover, we have previously examined how the amino acid difference in position 573 between albumin species (albumin from human, chimpanzee and orangutan have a lysine in this position, while all remaining species have a proline) affect FcRn binding. Replacing P573 in mouse albumin to lysine resulted in reduced binding to both mouse and human FcRn, while mutating K573 in HSA to proline gave rise to improved binding to hFcRn as well as towards monkey and mouse FcRn (Andersen et al., 2014, Nilsen et al., 2018).

In the current study we have focused on the human interaction and how the absence of L585 in HSA affect binding to hFcRn. However, whether the lack of L585 in HSA have the same negative effect on binding to FcRn from other species, especially from species that express albumin without the C-terminal leucine will be very interesting to address in future studies.

• Discussion lines 273-275: Report the level of normal HSA that lacks L585 in your experiments here.

We thank the reviewer for the suggestion and have accordingly added the following sentence to the discussion (Page 12, lines 276-277): *“In line with this, we found that about 3% of the total albumin in plasma from a healthy individual lacked the leucine.”*

• Discussion lines 296-298: The alternative point - C-term fusion of a drug to HSA might not be favorable if enzymatic cleavage occurs when the drug is fused.

We thank the reviewer for this comment, and have now included this alternative point as part of the discussion (Page 13, lines 299-300):

“Instead, genetic fusion of a drug of interest to the C-terminal of HSA may be favorable to prevent enzymatic cleavage, as long as the drug does not represent a substrate for CPA itself or compromise hFcRn binding.”

• Discussion lines 301-304: Has this K573P variant been in the clinic? Is there any evidence of immunogenicity?

The K573P variant has not yet been evaluated in humans, however there were no signs of immunogenicity following injection into mice transgenic for human FcRn or non-human primates (Andersen et al., 2014).

REVIEWERS' COMMENTS:

Reviewer #1 (Remarks to the Author):

My comments have been satisfactorily addressed, with the minor comment that 'speed vaced' is not typically used as a verb. This could be reformulated to something like ' were centrifuged in a speed vacuum concentrator to...'

Reviewer #2 (Remarks to the Author):

The authors have nicely addressed my concerns.

Reviewer #4 (Remarks to the Author):

This manuscript by Nilsen et. al. investigated in depth the interaction between albumin and FcRn and demonstrates that albumin variants have reduced binding to FcRn, explaining their shorter plasma half-life. The authors conclude that this helps understand albumin homeostasis and design of albumin therapeutics with long half-lives, an area of immediate relevance for folks in the field, and the scientific contributions are considerable.

The authors did a great job responding to my comments and revising the manuscript accordingly. I very much appreciated them marking the changes in the text for clarity and ease of re-review. I recommend publication and look forward to reading future publications from this group of scientists.